# The Mediating Role of Brain Structural Imaging Markers in Connecting Adverse Childhood Experiences and Psychological Resilience

**DOI:** 10.3390/children10020365

**Published:** 2023-02-11

**Authors:** Yun-Hsuan Chang, Meng-Heng Yang, Zai-Fu Yao, Meng-Che Tsai, Shulan Hsieh

**Affiliations:** 1Institute of Gerontology, College of Medicine, National Cheng Kung University, Tainan 70101, Taiwan; 2Institute of Behavioral Medicine, College of Medicine, National Cheng Kung University, Tainan 70101, Taiwan; 3Department of Psychology, National Cheng Kung University, Tainan 70101, Taiwan; 4Institute of Genomics and Bioinformatics, College of Life Sciences, National Chung Hsing University, Taichung 40227, Taiwan; 5College of Education, National Tsing Hua University, Hsinchu City, 30013, Taiwan; 6Research Center for Education and Mind Sciences, National Tsing Hua University, Hsinchu City 30013, Taiwan; 7Basic Psychology Group, Department of Educational Psychology and Counseling, National Tsing Hua University, Hsinchu City 30013, Taiwan; 8Department of Kinesiology, National Tsing Hua University, Hsinchu City 30013, Taiwan; 9Department of Pediatrics, National Cheng Kung University Hospital, College of Medicine, National Cheng Kung University, Tainan 70101, Taiwan; 10Department of Medical Humanities and Social Medicine, College of Medicine, National Cheng Kung University, Tainan 70101, Taiwan; 11Institute of Allied Health Sciences, National Cheng Kung University, Tainan 70101, Taiwan; 12Department of Public Health, College of Medicine, National Cheng Kung University, Tainan 70101, Taiwan

**Keywords:** adverse childhood experiences, multimodality neuroimaging, resilience, social resources

## Abstract

The impact of adverse childhood experiences (ACEs) on brain structure has been noticed. Resilience has been considered a protective characteristic from being mentally ill; however, the link between ACEs, psychological resilience, and brain imaging remains untested. A total of 108 participants (mean age 22.92 ± 2.43 years) completed the ACEs questionnaire and the Resilience Scale for Adults (RSA), with five subscales: personal strength (RSA_ps), family cohesion (RSA_fc), social resources (RSA_sr), social competence (RSA_sc), and future structured style (RSA_fss), and Magnetic Resonance Imaging (MRI) to acquire imaging data, and the fusion-independent component analysis was employed to determine multimodal imaging components. The results showed a significantly negative association between ACE subscales and RSA_total score (*ps* < 0.05). The parallel mediation model showed significant indirect mediation of mean gray matter volumes in the regions of the middle frontal gyrus, superior frontal gyrus, posterior cingulate, superior temporal gyrus, middle temporal gyrus, postcentral gyrus, middle temporal gyrus, and precuneus between childhood maltreatment and RSA_sr and RSA_sc. (*ps* < 0.05). This study highlighted the ACEs effect on gray matter volumes in the regions of the middle frontal gyrus, superior frontal gyrus, posterior cingulate, superior temporal gyrus, middle temporal gyrus, postcentral gyrus, middle temporal gyrus, and precuneus leading to decreased psychological resilience.

## 1. Introduction

Young adulthood, as a continuum of childhood and adolescence, is a critical period for an individual in transition to a full maturity status, where young individuals gain more autonomy and need to adapt to physical and psychological challenges linked to later life tasks [1]. Identifying the latent causes of adult health problems and addressing them promptly holds a potential utility that transforms the current diseased-based clinical practice into a proactive and preemptive behavior-centered care model [2]. Therefore, from an eco-bio-developmental perspective, personal traits may intertwine with environmental exposures and, in turn, shape overall health status because of the biological implication [3]. On the flip side, the prevention of long-term and adverse consequences is best managed by the protection provided by a stable and responsive microenvironment. This framework thus provides potential factors influencing how an individual’s experiences shape the origins of disparities in behavior, and health. Hence, the investigation into the role of protective factors (e.g., psychological resilience) in the face of adversity during the transition period of young adulthood will be crucial.

### 1.1. Childhood Adversity on Young Adult Development

Childhood adverse experiences have caught the attention of and have been recognized as a major health problem [4]. Childhood adversity or toxic stressors, now coined as adverse childhood experiences (ACEs), are conceptualized as contextual factors that not only confer strain on children and parents and undermine healthy functioning [5] but also result in excessive or prolonged activation of the physiologic stress response systems in the absence of the buffering protection [6]. The common types of ACEs could be broadly divided into acute and chronic stressors like a financial struggle, child maltreatment (emotional or physical), family dysfunction (domestic violence, parental divorce, substance abuse, criminal activity, or mental illness), and community/contextual traumatic events [2]. Trauma occurs only when children experience events or situations that surpass their coping abilities [7].

The current literature has summarized significant negative impacts of ACEs on physical and psychological health outcomes, such as alcoholism, drug abuse, depression, suicide, poor physical health, and obesity [8,9]. The shared mechanistic implications of these ACEs are founded on the plasticity of brain function that remains high throughout childhood and even young adulthood, as these changes brought about by stressful life events are likely to be programmed and embedded in the developing neural connections [10]. The disturbed neurophysiological functions may consequently lead to molecular and structural changes, particularly in young adults’ brains that are vulnerable to psycho-somatic and behavioral cues [11]. Additionally, the impact of ACEs may depend on the individual’s psychological resilience and ability to cope.

### 1.2. Role of Psychological Resilience Related to Childhood Adversity

Despite the negative impacts of ACEs, the emerging literature on resilience and counter-play of protective factors has provided a new perspective. The word “resilience” can be traced back to a 40-year longitudinal study (Kauai Longitudinal Study) in 1955, when the research team noticed that some children exposed to risk factors, such as low socioeconomic status, divorced family, and threatened by aggression, counterintuitively they did not develop maladaptive behaviors later in life [12]. Researchers have focused on strengths that individuals possess to help themselves overcome traumatic experiences [13,14,15]. Resilience, at its essence, refers to a mental process of negotiating, managing, and overcoming significant sources of stress or trauma [16]. Concerning the conceptualization of resilience, it may be seen as a personal trait representing an adaptive temperament or habitual effective coping strategy that aims at tackling stressful life events and coping with adversities from an outcome-oriented perspective [17,18,19]. To combat the detrimental effects of ACEs exerted on health outcomes, resilience, defined as “keeping calm and in control in the face of challenges,” was found to moderate the impact of ACE on grade repetition and poor school participation in a prior study [20]. In addition, several factors, such as living environment and positive individual personality, have been revealed and suggested as possible mediators in the relationship between increased exposure to ACEs and negative outcomes [21]. As such, psychological resilience is often regarded as a critical characteristic mediating the associations [20,22,23]. The synthesized analysis also suggested a dose relationship between ACE exposure and lower psychological resilience [24]. However, the way ACEs exposure is linked to psychological resilience, from a perspective of neuroscience, has not been explored. Thus, identifying resilience-related brain imaging features under ACE exposures may inform health policy mitigating the negative impacts of ACEs.

### 1.3. Childhood Adversity and Brain Imaging Markers

Previous studies have revealed a long-term and lasting change in neurophysiology and brain structure [25]. A positive correlation between negative childhood experiences and reduced global brain volume was reported, more particularly in the areas processing emotional stimuli and declarative memory, including the medial prefrontal lobe (mPFC), the insula, and the hippocampus. Smaller volumes were found in these areas for those who had been through childhood negative experiences compared to those who had not [26,27,28]. Oshri et al. (2019) reported that the more severe the ACE was encountered, the more reduced right amygdala volume was found in adults [29]. The reduction of basolateral subregions in the right amygdala is associated with an increased risk of anxiety, depression, and alcohol use. In addition, social economic status has also been noticed as an important factor and positively associated with limbic structure volume as well as with larger hippocampal volume [30]. Regarding social status, the greater resources received, the larger hippocampal volumes were noticed. Previous studies also demonstrate the effect of social status on the development of the prefrontal cortex and cognitive function [31,32].

Besides the changes in brain volumes, the effects of childhood emotional maltreatment (CEM) on the spontaneous amplitude of low-frequency fluctuations (ALFF) of neural activities in BOLD signals were investigated [33,34,35]. van der Werff et al. (2013) compared the adults with and without CEM on the limbic network, the default-mode network (DMN) and the salience network, and the left dorsomedial prefrontal cortex (dmPFC), and found the association between CEM with decreased ALFF in the dorsal anterior cingulate cortex (dACC), the precuneus as well as frontal regions [36]. The positive correlation between increased exposure to childhood violence and the greater reduced ALFF densities suggests that exposure to childhood violence is associated with neural network sparsity in adolescents [37]. Choi et al. (2012) compared the difference in the abnormalities in white matter (WM) tract integrity between a group with ACEs and healthy controls using diffusion tensor imaging (DTI). They found an impact of ACEs on altering fiber pathways and communicating adverse experiences to frontal, temporal, or limbic regions [38]. In addition, Ugwu et al. (2015) investigated the effects of ACEs, age, and gender on WM diffusion coefficients in tracts thought to be involved in emotion regulation in major depressive disorders (MDD) and healthy control (HC) individuals. They found a greater fractional anisotropy in the left hemisphere in the MDD group compared to the HC group [39]. It seems that the ACEs would alter the brain structure and cause negative consequences related to brain structure changes.

Recent studies have investigated the link between experiences-related structural [33,34,35] and functional connectivity changes in ACEs [36,40,41], providing either structural brain volume decreases or experience-related brain spontaneous brain activity changes associated with ACEs. For example, Yoshikawa et al. (2021), using diffusion tensor imaging to investigate the white matter integrity in autism spectrum disorder, have reported abnormal structural differences in the frontal brain regions [42]. However, a single neuroimaging technique (or data) cannot solely explain brain dynamics. To provide a holistic view of brain mechanisms underlying adverse childhood experience and experience-related joint brain imaging markers change, we adopted fusion approaches by Calhoun et al. (2009) to model imaging data with different types of informatics (e.g., spatial and temporal resolution scales) [43]. These methods use multiple imaging characteristics together to benefit from the combined information from unimodal image data. In addition, although previous studies have reported the mediating or moderating role of psychological resilience in the association between ACEs and negative consequences (i.e., poor academic outcome, mental illness, social costs) [17,18,19], the brain alteration seems to be affected by the ACEs exposures [44]. Hence, we sought to incorporate multimodal brain imaging markers in the mechanism underlying ACEs and psychological resilience.

### 1.4. The Present Study

As portrayed above, the present study based on the eco-bio-developmental framework was aimed toward examining the relationship between ACEs, psychological resilience, and brain functionality using a multimodal neuroimaging approach. Specifically, we tested the inverse association between ACEs burden and psychological resilience, and we further investigated the mediating role of brain functionality in the aforementioned association. We hypothesize that ACEs may be related to the changes in brain structure and/or function that are linked to psychological resilience in young adulthood. Please note, although previous studies have investigated neural correlates of ACEs and/or psychological resilience, to the authors’ knowledge, none has yet utilized a multimodal neuroimaging approach to observe both brain structure and function concerning ACEs and psychological resilience. Therefore, one of the aims of this study is to fill this research gap.

## 2. Materials and Methods

### 2.1. Participants

We recruited 108 participants (57 males and 51 females) from southern and central Taiwan through the Internet and advertisements on bulletin boards. All participants were aged between 20–29 years old with a mean age of 22.92 ± 2.43 years (standard deviation, SD) and educational years of 16.28 ± 1.80. All participants were given a written informed consent form approved by the Research Ethics Committee (REC, NCKU No. 109-419) and Institute of Review Board (IRB, JA-109-95) of Jen-Ai Hospital and signed to agree to participate in this study. After the completion of MRI scans and questionnaires, all participants received 2200 New Taiwan Dollars (NTD).

### 2.2. Resilience Score Measurement

The Resilience Scale for Adults (RSA) is a self-report questionnaire that was developed by Friborg et al. (2003) [45] and then revised in 2006. The Chinese version of RSA was translated by Wang (2007) [46]. The RSA contains 29 items, and each item is scored from 0 to 7, a higher score indicating greater resilience. The Chinese version has received great reliability of 0.89 based on the Taiwanese population using exploratory factor analysis (EFA), and five subscales were as follows: (1) personal strength (RSA_ps), (2) family cohesion (RSA_fc), (3) social resources (RSA_sr), (4) social competence (RSA_sc), and (5) future structured style (RSA_fss). The internal consistency for each subscale was 0.92, 0.85, 0.85, 0.83, and 0.87, respectively.

### 2.3. Adverse Childhood Experiences

To tailor the local social context, a modified version of the Adverse Childhood Experience International Questionnaire (ACE-IQ) was developed to assess the ACE type and frequency of the participants. The tool included a total of 24 items that can subdivide into childhood maltreatment (CM, 6 items), peer violence (PV, 4 items), family function (FF, 8 items), and environment safety (ES, 8 items). On the questionnaire, we asked the participants to rate the frequency and perceived impacts of each ACE item on a 4-point Likert-like scale from 0 (none) to 3 (always; enormously) if the ACE type was present [47]. For the analytic purpose, a sum score of the overall ACE questionnaire and its subscale was calculated and used to represent the severity of ACEs. For each dimension of ACE, the subscale was calculated by weighting the perceived impact with frequency. The Cronbach’s alpha for the total score was 0.52.

### 2.4. Image Acquisitions

The magnetic resonance imaging (MRI) data were acquired by a General Electronic (GE) MR750 3T scanner (GE Healthcare, Waukesha, WI, USA) in the Mind Research Imaging Center at National Cheng Kung University, Tainan, Taiwan. T1-weighted structural images with high resolution were obtained with fast corrupted gradient recalled echo sequence including 166 axial slices (TR/TE/flip angle 7.6 ms/3.3 ms/12°; the field of view [FOV] 22.4 × 22.4 cm^2^; matrices 224 × 224; slice thickness 1 mm). The entire process lasted for 3 min 38 s.

As for the resting-state functional images acquisition, an interleaved T2 *-weighted gradient-echo planar imaging (EPI) pulse sequence was used (TR/TE/flip angle = 2000 ms/30 ms/77°; matrices = 64 × 64; FOV = 22 × 22 cm^2^; slice thickness = 4 mm; voxel size = 3.4375 × 3.4375 × 4 mm). There were 245 volumes acquired (The first five dummy scans will be automatically discarded by the protocol to bring the magnetization system to a steady state), covering each participant’s entire brain. Participants were instructed to stay awake with eyes open to fixate on a white cross on a monitor screen during the resting-state functional scans. The entire scanning process lasted for 8 min and 10 s per participant (i.e., [number of samples + number of dummy scans] × TR = [240 + 5] × 2 = 490 s).

As for the diffusion tensor imaging (DTI) data acquisition, a spin-echo-echo planar sequence was used (TR/TE = 5500 ms/62–64 ms; 50 directions with b = 1000 s/mm^2^; 100 × 100 matrices; slice thickness = 2.5 mm; voxel size = 2.5 × 2.5 × 2.5 mm; the number of slices = 50; FOV = 25 cm; NEX = 3). Reverse DTI data was also acquired for preprocessing and correcting susceptibility-induced image distortions (i.e., top-up implemented in FSL). The acquisition parameters for the reverse DTI images matched those for the DTI acquisition, except for acquiring only six directions to avoid participant fatigue.

### 2.5. Image Preprocessing

#### 2.5.1. Structural MRI (sMRI)

We extracted structural images from the brain by the BET (Brain Extraction Tool) function implemented in FSL [48]. The -N option was chosen because the image contains most of the neck, and Voxel-based morphology (VBM) was used to characterize individual brain structural differences [49]. Next, tissue-type segmentation was carried out using FASTv4.0 [50], while the gray matter (GM) partial volume images were registered to the GM ICBM-152 template [51] using the non-linear registration tool FNIRT [52].

#### 2.5.2. Resting-State Functional MRI (rfMRI)

We applied the CONN toolbox 18a (www.nitrc.org/projects/conn, accessed on 20 October 2022) and SPM 12 (http://www.fil.ion.ucl.ac.uk/spm, accessed on 20 October 2022) of Matlab (The MathWorks, Inc., Natick, MA, USA) to preprocess the function images. For detailed parameters and procedures, please refer to Hsieh et al., 2021 [53]. After preprocessing the images, we calculated an index reflecting the intensity of regional spontaneous brain activity called the amplitude of low-frequency fluctuations (ALFF) [54]).

#### 2.5.3. Diffusion MRI (dMRI)

Diffusion tensor imaging (DTI), derived from preprocessed diffusion-weighted imaging (DWI) data [55] and has become the most widely used diffusion MRI method for extracting white matter tissue properties and identifying major white matter tracts. The FMRIB Software Library (FSL v5.0.9; www.fmrib.ox.ac.uk/fsl, accessed on 20 October 2022) was used for processing and analyses of the diffusion-weighted imaging (DWI) data. DTIFIT function [49] was applied to fit a tensor model at each voxel. Then TBSS in FSL [56,57] was used to perform a tract-based investigation of the DTI measurements. Finally, we acquired RD images for each participant. For detailed parameters and procedures, please refer to Hsieh et al., 2021 [53].

#### 2.5.4. Joint ICA Analysis

We have re-segmented the file prepared in the previous step into a 91 × 109 × 91 matrix with a voxel size of 2 × 2 × 2 mm and smooth using an isotropic Gaussian kernel with full width at half maximum (FWHM) sigma 8 mm. These files will then be analyzed with Fusion ICA Toolbox (FIT, http://mialab.mrn.org/software/fit/index.html, accessed on 20 October 2022). The analytics procedures of the fusion approach were adopted that similar to a previous study by Yang M.H. 2019 [58].

Each participant’s three-dimensional image was transformed into a single row and combined separately to create a matrix with dimensions [participant count] × [voxel count] for each imaging property. After normalization, the modified minimum description length (MDL) criteria is used to estimate the number of components. We chose MDL = 64 for the following analysis.

The data dimensionality was reduced by principal component analysis. The reduced feature matrix is decomposed into subject-specific mixing (loading) parameters and maximum independent component images by the informax algorithm. We employed ICASSO to conduct the ICA algorithm five times and identify the component with a stability index greater than 0.9.

In the subsequent correlation analysis, mixing parameters are used to evaluate the association of each component with RSA and ACE. The three modality images of each component are z-transformed separately. The values of all voxels in the spatial maps will be normalized to a mean of 0 and an SD of 1. Thus, each value, whether positive or negative, will be interpreted as its relative position in the contribution distribution. The threshold value for the spatial map is |Z| > 2.5. To provide more accurate cluster information, sMRI spatial maps are first converted from MNI coordinates to Talairach coordinates, which facilitated labeling and calculation of the cluster’s position and size according to the anatomical labels.

### 2.6. Statistics

#### 2.6.1. Correlation Analysis

Neither significant association between ACE, RSA, and the joint ICA components was correlated with age, nor educational level was found in the current study; partial correlation controlling for gender was performed to test the association among ACE, RSA, and the joint ICA components, using the Statistical Program for Social Sciences, version 22.0 (SPSS, Inc., Chicago, IL, USA, accessed on 20 October 2022). Priority selection of ICs with high relevance in both A and B paths. We then converted the positive and negative clusters of these components into brain masks; therefore, six masks, gray-matter volumes negative clusters (GMV_neg), gray-matter volumes positive clusters (GMV_pos), radial diffusivity negative clusters (RD_neg), radial diffusivity positive clusters (RD_pos), the amplitude of low-frequency fluctuations negative clusters (ALFF_neg), and amplitude of low-frequency fluctuations positive clusters (ALFF_pos) were obtained. We further used these masks to extract the brain signals of each subject (across all imaging modalities). Six measures were defined as independent variables.

#### 2.6.2. Mediation Analysis

Mplus analysis software version 8 was used to build a mediation path model with maximum likelihood estimation and bootstrapping methods. In addition, a bias-corrected method with the percentile bootstrap estimation approach was applied. The 95% confidence interval (CI) was estimated after 1000 bootstrap iterations, and the accepted hypothesis was accepted when a zero was not included in the values between the confidence intervals (CI) of the lower (LLCI) and upper bounds (ULCI). The data were transformed into z-scores before entering into the model, and all six brain measures were simultaneously considered in the model.

## 3. Results

### 3.1. Demographic Data

The internal consistency (Cronbach’s alpha) of the RSA scale and ACE was 0.89 and 0.52, respectively. The mean score of RSA total (140.4 ± 23.46) and subscales of RSA_ps, RSA_fc, RSA_sr, RSA_sc and RSA_fss are 27.48 ± 6.56, 33.66 ± 7.31, 42.36 ± 8.53, 18.69 ± 5.01 and 18.21 ± 5.1, respectively. The Cronbach’s alpha for RSA total score was 0.913, and for each subscale was 0.815, 0.806, 0.884, 0.828, and 0.830 in the current study.

### 3.2. Correlation between Adverse Childhood Experiences and Resilience for Adults

The correlation matrix showed a significantly negative association between RSA total score and ACE subscales, including childhood maltreatment and peer violence (r = −0.356 and r = −0.319, respectively). Regarding the subscales of RSA, the results showed a significantly negative correlation between RSA_ps, RSA_fc, and RSA_sr with childhood maltreatment (ACE_CM) and peer violence (ACE_PV) (Table 1). The subscale, ACE_FF, was found only negatively associated with RSA_fc.

### 3.3. Independent Components (ICs)

Of the 64 components, 29 remained because of their stability exceeding 0.9. However, 9 of the 29 components were rejected due to their presence of noticeable artifacts, such as sharp edges at the brain boundary or in the cerebrospinal fluid (CSF) area. Therefore, only 20 IC components remained for subsequent correlation analysis with RSA and ACE. The results showed that IC#16 (r = −0.343, *p* < 0.0001) was significantly correlated with ACE_ES. IC#27 (r = 0.213, *p* = 0.027) was significantly correlated with ACE_CM, and IC#31 also showed a tendency of positive association with ACE_CM (r = 0.179, *p* = 0.066). In addition, the results showed that IC#21 significantly correlated with RSA total score (r = 0.200, *p* = 0.039). IC#20 (r = 0.196, *p* = 0.043) and IC#25 (r = 0.243, *p* = 0.012) were significantly correlated with RSA_fc. IC#21 (r = 0.271, *p* = 0.005) and IC#31 (r = −0.230, *p* = 0.017) were significantly correlated with RSA_sr. IC#31 (r = −0.191, *p* = 0.049) was significantly correlated with RSA_sc.

### 3.4. Mediation Analysis

According to the intersection of both correlation analyses, IC#31 was selected for further parallel mediation analyses (Figure 1). The results showed a significant indirect effect between ACE_CM and RSA subscales mediating by the GM volumes of the IC#31 GMV_neg cluster.

### 3.5. The Mediator between RSA Subscales and ACE_CM

The main clusters of IC#31 GMV_neg are in the middle frontal gyrus, superior frontal gyrus, posterior cingulate, superior temporal gyrus, middle temporal gyrus, postcentral gyrus, middle temporal gyrus, and precuneus (Table 2). ACE_CM indirectly influences RSA subscales through these brain regions. The following diagram (Figure 2) shows the cluster of IC31’s GMV_neg spatial map after thresholding by |Z| > 2.5.

## 4. Discussion

This study sought to determine the multimodal neural features mediating the association between ACE and psychological resilience. A fusion joint ICA approach with multimodalities (rMRI, rfMRI, and dMRI) was conducted to investigate the association among multimodal imaging measures, ACEs, and resilience. Hypothetically, the mediation effect of neural features in the relationship between ACE and social domains of resilience (RSA_sr & RSA_sc) was investigated.

The results of the jICA analysis observed that one fusion multimodal imaging component (IC#31) spanning over the middle frontal gyrus, superior frontal gyrus, posterior cingulate, superior temporal gyrus, middle temporal gyrus, postcentral gyrus, middle temporal gyrus, precuneus, was significantly associated with resilience measured by RSA, especially with the subscales related to social domains (social resources, RSA_sr, and social competence, RSA_sc) (see Figure 2 and Table 2). The brain structures (i.e., gray matter volumes) in this component (IC#31) were further found to mediate the relationship between ACE and resilience (see Figure 2), suggesting the gray matter volumes other than the white matter or functional activity played a major role in mediating ACE and the social aspects of the resilience (see Figure 2).

Literature has shown that reduced brain volumes in the medial prefrontal lobe, the insula, and the hippocampus [31] were associated with adverse childhood experiences. In the current study, we further found that the component (IC#31) mediates the association between ACEs (impact of childhood maltreatment) and social domains of resilience in young adults. In addition, Oshri et al. (2019) also reported that the more severe the ACE was encountered, the more reduced right amygdala volume was found in adults [29]. Moreover, the reduction of basolateral subregions in the right amygdala is associated with an increased risk of anxiety, depression, and alcohol use. This effect may reflect the impact of childhood maltreatment on brain structural development, more specifically in the gray matter which is related to cognitive performance [59,60], and that indirectly affects resilience during young adulthood.

Luby et al. (2019) conducted a longitudinal study investigating the ACE effect with timing in the association with caregiver support and brain development. They found a pattern of the association between ACEs and caregiver support related to the development of the hippocampus and amygdala. With maternal support, positive associations with the insula, hippocampus, and amygdala volumes were found with school-age support. This developmental specificity in the association of psychosocial support to brain development regionally away from the impact of childhood maltreatment experiences [59]. The mediated effect of brain volumes indirectly correlated to ACE-CM and RSA_sr and RSA_sc supports the previous finding that the greater impact of maltreatment received during childhood correlates to lower volumes of brain regions spread wildly around the prefrontal cortex linking to the limbic system in an agreement to the previous report [61]. Moreno-López et al. (2020) conducted a comprehensive review discussing the brain structures and function in the association between resilient adults with histories of childhood maltreatment. They found a greater reduction in mPFC GMV in resilient adults and also reported reduced functional activity (FA) in tracts associated with the anterior thalamic radiation around the middle frontal gyrus, the right inferior frontal-occipital fasciculus, and the inferior longitudinal fasciculus and precuneus [62]. The categorized childhood maltreatment within ACEs may be associated with a distinct brain-behavior relationship in young healthy adults [63].

Common early life stressors include physical abuse, sexual abuse, emotional abuse, verbal abuse, neglect, social deprivation, disaster, family dysfunction, parental separation, or illness, which were categorized as ACEs. Much research has focused on the adult outcomes following childhood abuse and neglect, and in the current finding, childhood maltreatment was found as the more specific domain of ACEs affected by the brain structures. Literature has indicated that early life stress (ELS) is exposure during childhood to single or multiple events that exceed the child’s coping resources, resulting in prolonged periods of stress [64]. A recent study examining the relationship between reported ELS and brain morphology using magnetic resonance imaging (MRI) in a large sample of adults without a history of psychopathology showed similar Compared with those who experienced significant ELS, there were volumetric differences in brain structure, more reductions in the anterior cingulate cortex and caudate nucleus, in those experienced the least ELS [26]. These results may imply the adverse effect of early life events of ACEs on brain integrity. The size of the prefrontal cortex increases slowly until age 8, followed by rapid growth throughout adolescence [65]. Thus, Barker et al. examined the relationship between brain morphology and age at early life stress presentation in a sample of healthy individuals with no history of psychopathology or brain disease [66]. They examined the regions, including those involved in post-traumatic stress disorder and other emotional behaviors, including the hippocampus, amygdala, anterior cingulate cortex, insula, and caudate nucleus, and found that the later developmental stages of ACEs onset are associated with greater volume loss in these regions. The studied population in the current study were young adults (20–29 years old), and the ACEs reports contain the events that happened before their 18s, which could be more specified into early and late events in further analysis to compare the impact of ACEs on brain structures. Because the effects of early ACEs on brain integrity are related to the age at which ACEs occur, deleterious effects only associated with exposure in late childhood/adolescence were suggested.

### Limitations and Future Directions

There are some limitations. First, the collection of data is cross-sectional at its best, although the ACEs are defined by the occurrence before the age of 18 years that is precedent to the acquisition of brain images and the measured status of resilience. The effects mentioned in this paper should be interpreted as statistical estimates of the associations rather than causal impacts. Also, ACEs were self-reported and thus may be subject to recall bias. Second, as our statistical analyses were exploratory, the results were not corrected for multiple comparisons. The selection of the networks of interest was data-driven. Because very few studies have been dedicated to testing the causal relationship between ACEs and longitudinal changes in these networks in youth, future research of larger participation may be needed to replicate our findings. Third, the participants were generally healthy young adults, and whether the findings could be generalized to those with more severe traumas or from a more deprived community may also require more research to validate. Although trauma is one possible outcome of exposure to adversity, a traumatic event is not an isolated event viewed equally by those who experience it. What an adult perceives as traumatic may be very different from what a child perceived as traumatic. It would be better to conduct a longitudinal design study to clarify the different impacts of ACEs and traumatic events in the future.

Based on the current findings of the indirect mediation effect of the middle frontal gyrus, superior frontal gyrus, posterior cingulate, superior temporal gyrus, middle temporal gyrus, postcentral gyrus, middle temporal gyrus, and precuneus in between the ACEs and psychological resilience in young adults, the incidence and long-term effects of ACEs in adults should be considered in future studies. In addition, the importance of social support was indicated as an important factor in enhancing psychological resilience, which is supportive of previous findings [67], the more specific dimension of social resources related to psychological resilience could be conducted in future studies to understand the long-term effects further.

## 5. Implications and Conclusions

The current study highlights the multimodal neuroimages in the study of the association between adverse childhood experiences and resilience in young adults. From the psycho-pathological standpoint, a key result of this study is to demonstrate the impact of ACEs, more specifically childhood maltreatment, on the brain structures, and the gray matter volumes. The apparent association between maltreatment impact and resilience for young adults is mediated by gray matter related to cognitive performance. Moreover, the current study further found an indirect association between childhood maltreatment of ACE impact and social domains of resilience for young adults that is mediated by the middle frontal gyrus, superior frontal gyrus, posterior cingulate, superior temporal gyrus, middle temporal gyrus, postcentral gyrus, middle temporal gyrus, and precuneus Potentially, this finding could play as a fundamental neuro-basis mechanistic chain that forms the pathway from adverse childhood experiences to resilience.

## Figures and Tables

**Figure 1 children-10-00365-f001:**
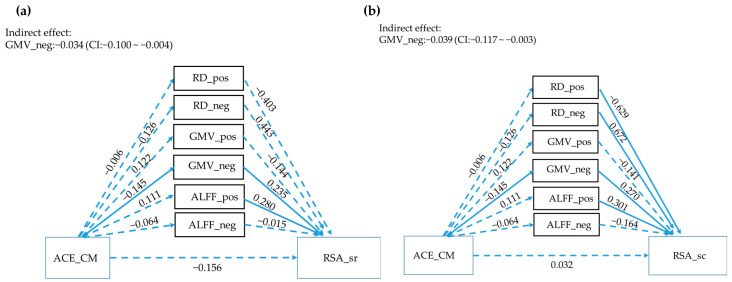
Parallel mediation analyses of ACEs, multimodality imaging, and RSA. Solid lines represent significant paths (*p* < 0.05), while the dashed line represents non-significant paths. (*p* > 0.05). Path values represent standardized β weights and *p* values. CI represents the 1000 samples bootstrapped with 95% confidence intervals for the indirect effects and total effects. GMV = gray-matter volumes, ALFF = amplitude of low-frequency fluctuations; RD = Radial diffusivity; pos = positive effect; neg = negative effect. ACEs: Adverse Childhood Experiences questionnaire; ACE_CM: Childhood Maltreatment domain of ACE; ACE_PV: Peer Violence domain of ACE; ACE_FF: Family Function domain of ACE; RSA: Resilience Scale for Adults; RSA_total: total score of RSA; RSA_ps: personal strength domain of RSA; RSA_fc: family cohesion domain of RSA; RSA_sr: social resources domain of RSA; RSA_sc: social competence of RSA; RSA_fss: future structured style of RSA. (**a**) the upper figure shows the different modalities mediating the correlation between ACEs and RSA_sr; (**b**) the below figure shows the different modalities mediating the correlation between ACEs and RSA_sc.

**Figure 2 children-10-00365-f002:**
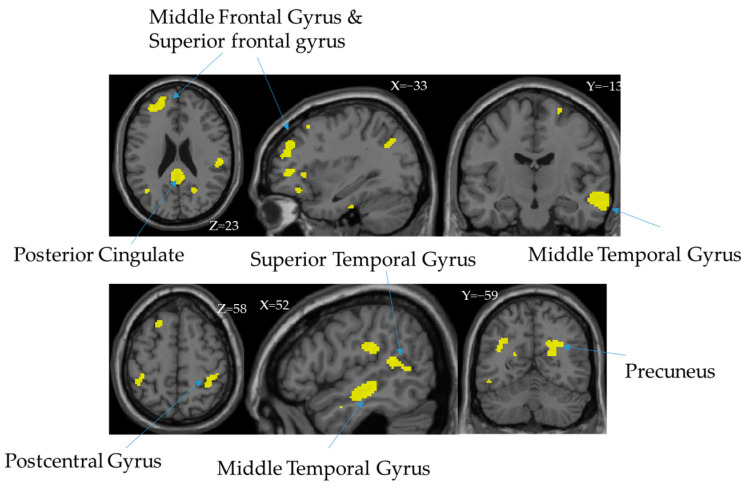
Conceptual illustration to present possible relation between brain regions and RSA.

**Table 1 children-10-00365-t001:** Correlation matrix between Adverse Childhood Experiences and Resilience Scale for Adults.

	ACE_CM		ACE_PV		ACE_FF		ACE_ES	
	r	p	r	p	r	p	r	p
RSA_total	−0.356	0.000	−0.319	0.001	−0.109	0.265	0.013	0.897
RSA_ps	−0.256	0.008	−0.213	0.029	−0.013	0.897	−0.039	0.689
RSA_fc	−0.501	0.000	−0.276	0.004	−0.323	0.001	−0.013	0.896
RSA_sr	−0.240	0.013	−0.341	0.000	−0.019	0.843	0.047	0.629
RSA_sc	−0.064	0.516	−0.239	0.013	−0.077	0.431	0.007	0.946
RSA_fss	−0.132	0.178	−0.002	0.986	0.087	0.375	0.043	0.659

Note: ACE: Adverse Childhood Experiences questionnaire; ACE_CM: Childhood Maltreatment domain of ACE; ACE_PV: Peer Violence domain of ACE; ACE_FF: Family Function domain of ACE; RSA: Resilience Scale for Adults; RSA_total: total score of RSA; RSA_ps: personal strength domain of RSA; RSA_fc: family cohesion domain of RSA; RSA_sr: social resources domain of RSA; RSA_sc: social competence of RSA; RSA_fss: future structured style of RSA.

**Table 2 children-10-00365-t002:** The correlation matrix between independent components (ICs) and the RSA (Resilience Scale for Adults) subscales, as well as the ACE (Adverse Childhood Experiences) subscales.

Component No.	ACE_CM		ACE_PV		ACE_FF		ACE_ES		RSA total		RSA_ps		RSA_fc		RSA_sr		RSA_sc		RSA_fss	
	r	p	r	p	r	p	r	p	r	p	r	p	r	p	r	p	r	p	r	p
1	−0.036	0.709	−0.106	0.278	0.041	0.675	0.070	0.476	−0.008	0.938	0.029	0.767	−0.089	0.364	0.018	0.852	−0.033	0.736	0.057	0.559
4	−0.121	0.216	0.041	0.679	−0.028	0.776	0.125	0.201	0.067	0.492	0.059	0.544	0.095	0.331	0.108	0.268	−0.068	0.490	−0.017	0.859
7	−0.003	0.975	0.127	0.194	−0.02	0.838	−0.021	0.832	0.109	0.262	0.111	0.255	0.108	0.268	0.002	0.982	0.098	0.313	0.104	0.288
10	−0.137	0.158	0.059	0.544	−0.03	0.758	0.008	0.934	−0.02	0.838	−0.105	0.281	0.062	0.529	0.046	0.639	−0.06	0.538	−0.062	0.530
12	−0.007	0.942	0.090	0.359	−0.117	0.232	0.175	0.071	0.029	0.766	−0.039	0.689	0.092	0.344	0.018	0.851	0.073	0.455	−0.051	0.605
14	−0.009	0.926	−0.052	0.593	−0.067	0.492	0.047	0.627	0.11	0.261	0.102	0.297	0.062	0.525	0.053	0.591	0.145	0.137	0.054	0.582
15	−0.046	0.637	−0.086	0.379	−0.036	0.715	0.06	0.536	0.036	0.71	0.104	0.284	−0.006	0.949	0.034	0.729	−0.029	0.769	0.013	0.895
16	−0.026	0.793	0.047	0.630	−0.118	0.227	−0.343	0.000	−0.028	0.774	−0.062	0.524	−0.005	0.959	−0.03	0.758	0.048	0.621	−0.039	0.693
17	−0.088	0.367	−0.018	0.855	0.013	0.893	0.052	0.592	0.108	0.267	0.137	0.161	0.067	0.494	0.069	0.479	−0.003	0.978	0.114	0.243
19	0.134	0.169	0.065	0.506	0.121	0.215	−0.016	0.871	0.023	0.813	0.034	0.726	0.012	0.903	−0.01	0.922	0.006	0.952	0.055	0.574
20	−0.174	0.072	0.110	0.258	−0.074	0.448	−0.116	0.235	0.113	0.246	0.064	0.512	0.196	0.043	0.002	0.981	−0.014	0.886	0.169	0.083
21	−0.053	0.591	−0.091	0.353	0.015	0.875	0.048	0.622	0.200	0.039	0.112	0.251	0.054	0.582	0.271	0.005	0.142	0.144	0.111	0.259
22	0.083	0.393	0.042	0.665	0.099	0.308	0.021	0.829	0.001	0.990	0.006	0.949	−0.065	0.504	0.043	0.662	0.048	0.627	−0.026	0.794
23	0.043	0.662	0.139	0.154	−0.112	0.252	0.012	0.905	0.010	0.921	−0.059	0.545	0.138	0.155	−0.028	0.772	−0.022	0.818	−0.009	0.924
24	−0.096	0.325	−0.044	0.653	−0.086	0.381	−0.003	0.972	0.043	0.662	0.024	0.808	0.091	0.351	0.047	0.631	−0.024	0.807	−0.020	0.841
25	−0.041	0.676	0.018	0.857	−0.019	0.85	0.063	0.522	−0.014	0.883	−0.071	0.468	0.243	0.012	−0.091	0.351	−0.118	0.226	−0.058	0.556
26	−0.063	0.518	−0.049	0.613	0.031	0.752	0.104	0.286	0.023	0.811	0.031	0.753	0.009	0.923	0.019	0.845	0.145	0.137	−0.119	0.224
27	0.213	0.027	0.071	0.469	0.10	0.303	−0.040	0.686	−0.112	0.251	−0.089	0.359	−0.118	0.225	−0.128	0.189	−0.006	0.949	−0.011	0.913
28	0.019	0.844	−0.100	0.305	−0.019	0.849	−0.090	0.356	−0.043	0.658	−0.051	0.602	−0.11	0.259	0.072	0.460	0.009	0.930	−0.104	0.290
31	0.179	0.066	0.024	0.807	0.079	0.419	−0.019	0.843	−0.172	0.076	−0.031	0.749	−0.055	0.575	−0.230	0.017	−0.191	0.049	−0.107	0.277

Note: ACEs: Adverse Childhood Experiences questionnaire; ACE_CM: Childhood Maltreatment domain of ACE; ACE_PV: Peer Violence domain of ACE; ACE_FF: Family Function domain of ACE; RSA: Resilience Scale for Adults; RSA_total: total score of RSA; RSA_ps: personal strength domain of RSA; RSA_fc: family cohesion domain of RSA; RSA_sr: social resources domain of RSA; RSA_sc: social competence of RSA; RSA_fss: future structured style of RSA.

## Data Availability

The datasets used and analyzed during the current study are available from the corresponding author upon reasonable request.

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
