# Peer review of "The Mediating Role of Brain Structural Imaging Markers in Connecting Adverse Childhood Experiences and Psychological Resilience"

_children, 2023, doi:10.3390/children10020365_

Round 1

Reviewer 1 Report

The article analyzes the relationship - adverse childhood experiences, gray matter volume and psychological resilience.

 I have the following recommendations for authors:

 1. I recommend that the abstract be rephrased to give the objectives, tasks, results and conclusion more clearly in a structured form. The abstract presented in this way is not informative enough.

2. The introduction is well presented. The authors need to better distinguish between the terms: traumatic events and psychological resilience. An analysis of the research conducted on ACEs and childhood trauma would be useful to further clarify the specifics of the research conducted.

3.Materials and methods. The presented data is well systematized. What is the educational level of the studied contingent? Are there differences and whether this is expected to affect the results given the brain plasticity data. Statistical methods are relevant to the tasks at hand.

4. Results. Detailed and informative and clearly arranged.

5.Discussion. I recommend discussing the age of childhood adverse events /early and later/and the relationship with neuroimaging outcomes.

6. I found no directions for future research.

 The reviewer

Reviewer 2 Report

The manuscript is really of great interest. It is well-structured and well-written.

I particularly appreciated the epistemological basis of the method, informed by the awareness that “a single neuroimaging technique (or data) cannot solely explain brain dynamics.”

The Results and the Discussion sections seem adequate to the research design and are consistent with the Methodology section.

Overall, it was a pleasure reading the manuscript!

I only notice below a few minor typos:

Line 91: please replace “;” with “,”.

Lines 97-99: something is missing here.

Lines 208-209: “Diffusion tensor imaging (DTI), derived from preprocessed diffusion-weighted im-208 aging (DWI) data [48] and has become the most widely used diffusion MRI method…”: something is not right with this sentence.

Line 216: “Hsieh 2021” is missing the reference number.

Line 254: double period.

Line 257: please capitalize “mediation”.

Reviewer 3 Report

First of all I would like to thank you for the opportunity to review this research. I consider that this research needs to be improved for publication. 

In line 91, the citation needs to be adapted to the journal's rules.

Also, the text should be right-justified.

Regarding the methodological section, I consider that the study is well thought out and justified, however I believe that the discussion needs to be extended and improved. 

Round 2

Reviewer 1 Report

The article has been revised. The remarks made have been taken into account and in this form the article can be submitted for publication.

The reviewer